# The mTOR pathway genes *MTOR*, *Rheb*, *Depdc5*, *Pten*, and *Tsc1* have convergent and divergent impacts on cortical neuron development and function

Lena H Nguyen[1,2]*, Youfen Xu[2], Maanasi Nair[2], Angelique Bordey[2]*

[1]Department of Neuroscience, School of Behavioral and Brain Sciences, University of Texas at Dallas, Richardson, United States; [2]Departments of Neurosurgery and Cellular & Molecular Physiology, Wu Tsai Institute, Yale University School of Medicine, New Haven, United States

*For correspondence:
lena.nguyen@utdallas.edu (LHN);
angelique.bordey@yale.edu (AB)

**Abstract** Brain somatic mutations in various components of the mTOR complex 1 (mTORC1) pathway have emerged as major causes of focal malformations of cortical development and intractable epilepsy. While these distinct gene mutations converge on excessive mTORC1 signaling and lead to common clinical manifestations, it remains unclear whether they cause similar cellular and synaptic disruptions underlying cortical network hyperexcitability. Here, we show that in utero activation of the mTORC1 activator genes, *Rheb* or *MTOR*, or biallelic inactivation of the mTORC1 repressor genes, *Depdc5*, *Tsc1*, or *Pten* in the mouse medial prefrontal cortex leads to shared alterations in pyramidal neuron morphology, positioning, and membrane excitability but different changes in excitatory synaptic transmission. Our findings suggest that, despite converging on mTORC1 signaling, mutations in different mTORC1 pathway genes differentially impact cortical excitatory synaptic activity, which may confer gene-specific mechanisms of hyperexcitability and responses to therapeutic intervention.

## eLife assessment

This manuscript examines shared and divergent mechanisms of disruptions of five different mTOR pathway genes on embryonic mouse brain neuronal development. The significance of the manuscript is **important**, because it bridges several different genetic causes of focal malformations of cortical development. The strength of evidence is **compelling**, relying on both gain and loss of function, demonstrating differential impact on excitatory synaptic activity, conferring gene-specific mechanisms of hyperexcitability. The results have both theoretical and practical implications for the field of developmental neurobiology and clinical epilepsy.

## Introduction

Focal malformations of cortical development (FMCDs), including focal cortical dysplasia type II (FCDII) and hemimegalencephaly (HME), are the most common causes of intractable epilepsy in children (*Harvey et al., 2008*; *Blumcke et al., 2017*). These disorders are characterized by abnormal brain cytoarchitecture with cortical overgrowth, mislamination, and cellular anomalies, and are often associated with developmental delay and intellectual disability (*Guerrini and Dobyns, 2014*). Early immunohistochemical studies identified hyperactivation of the mechanistic target of rapamycin complex 1 (mTORC1) signaling pathway in resected brain tissue from individuals with FCDII and HME, leading to

the classification of these disorders as 'mTORopathies' (*Crino, 2009*). More recently, somatic mutations in numerous regulators of the mTORC1 pathway were identified as the genetic causes of FCDII and HME (*Marsan and Baulac, 2018*; *Mühlebner et al., 2019*; *Gerasimenko et al., 2023*). Accumulating evidence shows that these mutations occur in a subset of dorsal telencephalic progenitor cells that give rise to excitatory neurons during fetal development, resulting in brain somatic mosaicism (*D'Gama et al., 2017*; *Chung et al., 2023*). These genetic findings provide opportunities for gene therapy approaches targeting the mTORC1 pathway for FCDII and HME.

mTORC1 is an evolutionarily conserved protein complex that promotes cell growth and differentiation through the regulation of protein synthesis, metabolism, and autophagy (*Laplante and Sabatini, 2012*). mTORC1 is composed of MTOR, a serine/threonine kinase that exerts the complex's catalytic function, Raptor, PRAS40, and mLST8. Activation of mTORC1 is controlled by a well-described upstream cascade involving multiple protein regulators (*Dibble and Cantley, 2015*). Stimulation by growth factors activates phosphoinositide 3-kinase (PI3K), which leads to the activation of phosphoinositide-dependent kinase 1 (PDK1) and subsequent phosphorylation and activation of AKT. Activated AKT phosphorylates and inactivates tuberous sclerosis complex 1/2 (TSC1/2 complex; consisting of TSC1 and TSC2 proteins), which releases the brake on Ras homolog enriched in brain (RHEB), a GTP-binding protein that directly activates mTORC1. This mTORC1-activating cascade is negatively controlled by the phosphatase and tensin homolog (PTEN) protein, which inhibits PI3K activation of PDK1. Additionally, mTORC1 signaling is regulated by a separate nutrient-sensing GAP Activity Towards Rags 1 (GATOR1) complex, which consists of a DEP domain containing 5 (DEPDC5) and the nitrogen permease regulator 2-like (NPRL2) and 3-like (NPRL3) proteins (*Bar-Peled and Sabatini, 2014*). The GATOR1 complex serves as a negative regulator of mTORC1 that inhibits mTORC1 at low amino acid levels.

Pathogenic mutations in numerous regulators that activate mTORC1 signaling, including *PIK3CA, PTEN, AKT3, TSC1, TSC2, MTOR, RHEB, DEPDC5, NPRL2,* and *NPRL3,* have been identified in HME and FCDII (*Chung et al., 2023*). Genetic targeting of these genes in mouse models consistently recapitulates the epilepsy phenotype, supporting an important role for these genes in seizure generation (*Nguyen and Bordey, 2021*; *Nguyen and Bordey, 2022*). However, while all these genes impinge on the mTORC1 pathway, many of them also participate in mTORC1-independent functions through parallel signaling pathways (*Nguyen and Bordey, 2021*; *Neuman and Henske, 2011*; *Chetram and Hinton, 2012*; *Lien et al., 2017*; *Manning and Toker, 2017*; *Swaminathan et al., 2018*), and it is unclear whether mutations affecting different mTORC1 pathway genes lead to cortical hyperexcitability through common neural mechanisms. In this study, we examined how disruption of five distinct mTORC1 pathway genes, *Rheb, MTOR, Depdc5, Pten,* and *Tsc1,* individually impact pyramidal neuron development and electrophysiological function in the mouse medial prefrontal cortex (mPFC). Collectively, we found that activation of the mTORC1 activator genes, *Rheb* and *MTOR,* or inactivation of the mTORC1 repressor genes, *Depdc5, Tsc1,* and *Pten,* largely leads to similar alterations in neuron morphology and membrane excitability but differentially impacts excitatory synaptic activity. The latter has implications for cortical network function and seizure vulnerability and may affect how individuals with different genotypes respond to targeted therapeutics.

## Results

### Expression of *Rheb*$^{Y35L}$, *MTOR*$^{S2215Y}$, *Depdc5*$^{KO}$, *Pten*$^{KO}$, and *Tsc1*$^{KO}$ leads to varying magnitudes of neuronal enlargement and mispositioning in the cortex

To model somatic gain-of-function mutations in the mTORC1 activators, we individually expressed plasmids encoding *Rheb*$^{Y35L}$ or *MTOR*$^{S2215Y}$, respectively, in select mouse neural progenitor cells during late corticogenesis, at embryonic day (E) 15, via in utero electroporation (IUE) (*Figure 1a and b*). These pathogenic variants were previously detected in brain tissue from children with FCDII and HME associated with intractable seizures (*Nakashima et al., 2015*; *Mirzaa et al., 2016*; *Møller et al., 2016*; *Baldassari et al., 2019*; *Zhao et al., 2019*; *Lee et al., 2021*). We specifically targeted a subset of late-born progenitor cells that generate excitatory pyramidal neurons destined to layer 2/3 in the mPFC to mimic frontal lobe somatic mutations and the genetic mosaicism characteristic of these lesions. To model the somatic loss-of-function mutations in mTORC1 repressors, we expressed

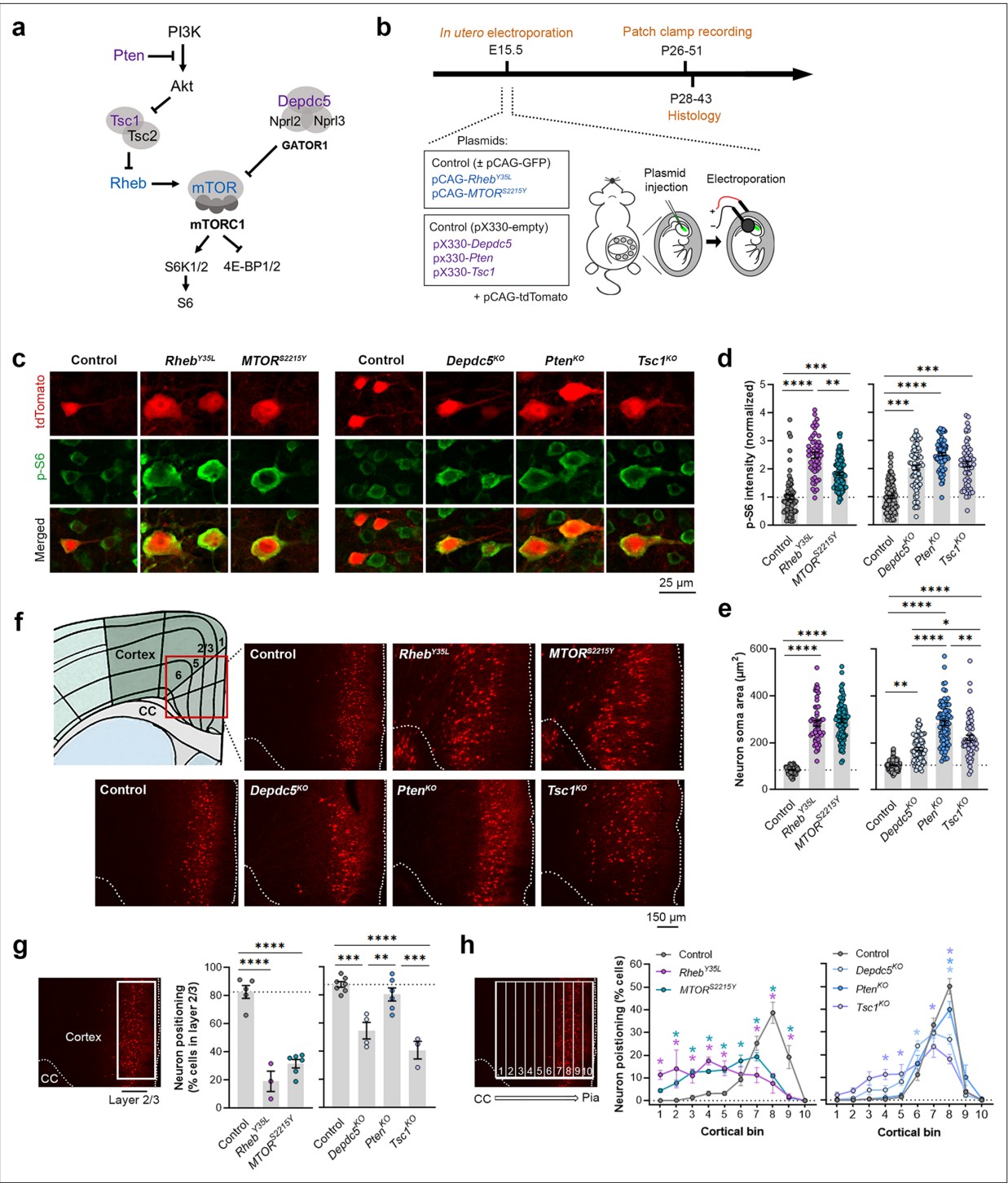

**Figure 1.** Expression of $Rheb^{Y35L}$, $MTOR^{S2215Y}$, $Depdc5^{KO}$, $Pten^{KO}$, and $Tsc1^{KO}$ leads to varying magnitudes of neuronal enlargement and mispositioning in the cortex. (**a**) Diagram of the PI3K-mTORC1 pathway. Activation of mTOR complex 1 (mTORC1) signaling is controlled by positive (blue) and negative (purple) regulators within the pathway. (**b**) Diagram of overall experimental timeline and approach. in utero electroporation (IUE) was performed at E15.5. A cohort of animals was used for patch clamp recording at P26-51 and another cohort was used for histology at P28-43. (**c**) Representative images of tdTomato+ cells (red) and p-S6 staining (green) in mouse medial prefrontal cortex (mPFC) at P28-43. (**d**) Quantification of p-S6 staining intensity (normalized to the mean control) in tdTomato+ neurons. (**e**) Quantification of tdTomato+ neuron soma size. (**f**) Representative images of tdTomato+ neuron (red) placement and distribution in coronal mPFC sections. Red square on the diagram denotes the imaged area for all groups. CC, corpus callosum. (**g**) Quantification of tdTomato+ neuron placement in layer 2/3. Left diagram depicts the approach for analysis: the total number of tdTomato+ neurons within layer 2/3 (white square) was counted and expressed as a % of total neurons in the imaged area. Right bar graphs show the quantification.

*Figure 1 continued on next page*

*Figure 1 continued*

(**h**) Quantification of tdTomato+ neuron distribution across cortical layers. Left diagram depicts the approach for analysis: the imaged area was divided into 10 equal-sized bins across the cortex, spanning the corpus callosum to the pial surface (white grids); the total number of tdTomato+ neurons within each bin was counted and expressed as a % of total neurons in the imaged area. Right bar graphs show the quantification. For graphs (**d, e**): n=4–8 mice per group, with 6–15 cells analyzed per animal. For graphs (**g, h**): n=3–7 mice per group, with 1 brain section analyzed per animal. Statistical comparisons were performed using (**d, e**) nested one-way ANOVA (fitted to a mixed-effects model) to account for correlated data within individual animals, (**g**) one-way ANOVA, or (**h**) two-way repeated measures ANOVA. Post-hoc analyses were performed using Holm-Šídák multiple comparison test. *p<0.05, **p<0.01, ***p<0.001, ****p<0.0001. All data are reported as the mean of all neurons or brain sections in each group ± SEM.

The online version of this article includes the following source data and figure supplement(s) for figure 1:

**Source data 1.** Summary statistics for *Figure 1*.

**Figure supplement 1.** Distribution of p-S6 staining intensity and neuron soma size among individual animals at P28-43.

**Figure supplement 2.** Neuron soma sizes at P7-9.

**Figure supplement 2—source data 1.** Summary statistics for *Figure 1—figure supplement 2*.

plasmids encoding previously validated CRISPR/Cas9 guide RNAs against *Depdc5* (*Ribierre et al., 2018*), *Tsc1* (*Lim et al., 2017*), or *Pten* (*Xue et al., 2014*) to individually knockout (KO) the respective genes using the same IUE approach (*Figure 1a and b*). As a control for the activating mutations, we expressed plasmids encoding fluorescent proteins under the same CAG promoter. As a control for the CRISPR/Cas9-mediated knockouts, we used an empty CRISPR/Cas9 vector containing no guide RNA sequences. To verify that the expression of these plasmids leads to mTORC1 hyperactivation, we assessed the phosphorylated levels of S6 (i.e. p-S6), a downstream substrate of mTORC1, using immunostaining in brain sections from postnatal day (P) 28–43 mice. As expected, we found that expression of the $Rheb^{Y35L}$, $MTOR^{S2215Y}$, $Depdc5^{KO}$, $Pten^{KO}$, and $Tsc1^{KO}$ plasmids all led to significantly increased p-S6 staining intensity, supporting that the expression of each of these plasmids leads to increased mTORC1 signaling (*Figure 1c and d*, *Figure 1—figure supplement 1a*, *Figure 1—source data 1*).

Considering the cytoarchitectural abnormalities associated with mTORC1 hyperactivation, we compared neuron soma size and positioning in the cortex following in utero activation of *Rheb* and *MTOR* and inactivation of *Depdc5*, *Pten*, and *Tsc1* in P28-43 mice. While all experimental conditions led to increased neuron soma size, the magnitude of the enlargement was dependent on the specific gene that was targeted (*Figure 1c and e*, *Figure 1—figure supplement 1b*, *Figure 1—source data 1*). In particular, expression of $Rheb^{Y35L}$ and $MTOR^{S2215Y}$ led to the largest soma size changes, with a >threefold increase from control. Expression of $Pten^{KO}$ led to a similarly large increase of 2.7-fold, while expression of $Depdc5^{KO}$ and $Tsc1^{KO}$ led to a 1.7 and 2.1-fold increase, respectively. The increase in the $Pten^{KO}$ condition was significantly higher than both the $Depdc5^{KO}$ and $Tsc1^{KO}$ conditions, and the increase in the $Tsc1^{KO}$ condition was significantly higher than the $Depdc5^{KO}$ condition. Although the above analysis was performed at P28-43, the enlargement of neuron soma sizes was already detected by P7-9 (*Figure 1—figure supplement 2a–c*). In terms of neuronal positioning, all experimental conditions, except for $Pten^{KO}$, resulted in neuron misplacement (*Figure 1f–h*, *Figure 1—source data 1*). The $Rheb^{Y35L}$ and $MTOR^{S2215Y}$ conditions led to the most severe phenotype: ~70–80% of the neurons were misplaced outside of layer 2/3 (*Figure 1g*), with the mispositioned neurons evenly scattered across the deeper layers (*Figure 1h*). For the $Depdc5^{KO}$ and $Tsc1^{KO}$ conditions, ~45–60% of the neurons were misplaced outside of layer 2/3 (*Figure 1g*), with the mispositioned neurons found closer to layer 2/3 (*Figure 1h*). Taken together, these studies show that the expression of $Rheb^{Y35L}$, $MTOR^{S2215Y}$, $Depdc5^{KO}$, $Pten^{KO}$, and $Tsc1^{KO}$ leads to varying magnitudes of neuronal enlargement and mispositioning in the cortex. Of note, $Pten^{KO}$ neurons, despite exhibiting a 2.7-fold increase in neuron soma size, were mostly correctly positioned in layer 2/3. These findings suggest that while all experimental conditions lead to increased soma size, not all lead to neuron mispositioning, suggesting defective migration and the subsequent impact on neuron positioning occur independently of cell size.

## Expression of *Rheb*$^{Y35L}$, *MTOR*$^{S2215Y}$, *Depdc5*$^{KO}$, *Pten*$^{KO}$, and *Tsc1*$^{KO}$ universally leads to decreased depolarization-induced excitability, but only *Rheb*$^{Y35L}$, *MTOR*$^{S2215Y}$, and *Tsc1*$^{KO}$ expression leads to depolarized resting membrane potential

To elucidate the contribution of each experimental condition to the function of cortical neurons, we obtained whole-cell patch clamp recordings of layer 2/3 pyramidal neurons at P26-P51 (*Figure 2a*). The *Rheb*$^{Y35L}$ and *MTOR*$^{S2215Y}$ conditions were compared to a control group expressing fluorescent proteins under the same CAG promoter. The *Depdc5*$^{KO}$, *Pten*$^{KO}$, and *Tsc1*$^{KO}$ conditions were compared to a CRISPR/Cas9 empty vector control. Recordings of control and experimental conditions were alternated to match the animal ages.

Consistent with the changes in soma size (*Figure 1c and e*), recorded neurons displayed increased membrane capacitance in all experimental conditions (*Figure 2b*, *Figure 2—figure supplement 1a*, *Figure 2—source data 1*). Neurons expressing the *MTOR*$^{S2215Y}$ variant had a larger capacitance increase than those expressing the *Rheb*$^{Y35L}$ variant. *Pten*$^{KO}$ and *Tsc1*$^{KO}$ neurons had similar increases in capacitance that were both larger than that of the *Depdc5*$^{KO}$ neurons. All neurons across the experimental conditions also had increased resting membrane conductance in a pattern that followed that of the capacitance (*Figure 2c*, *Figure 2—figure supplement 1b*, *Figure 2—source data 1*). However, while *Rheb*$^{Y35L}$, *MTOR*$^{S2215Y}$, and *Tsc1*$^{KO}$ expression led to depolarized resting membrane potential (RMP), *Depdc5*$^{KO}$ and *Pten*$^{KO}$ expression did not significantly affect the RMP (*Figure 2d*, *Figure 2—figure supplement 1c*, *Figure 2—source data 1*). To assess whether these changes impacted neuron intrinsic excitability, we examined the action potential (AP) firing response to depolarizing current injections. For all experimental conditions, neurons fired fewer APs for current injections above 100 pA compared to the respective control neurons (*Figure 2e and f*, *Figure 2—source data 1*). This decrease in intrinsic excitability is reflected in the increased rheobase (i.e. the minimum current required to induce an AP) in all experimental conditions, with the *MTOR*$^{S2215Y}$ and *Rheb*$^{Y35L}$ conditions leading to the largest rheobase increases (*Figure 2g*, *Figure 2—figure supplement 1d*, *Figure 2—source data 1*). Collectively, these findings indicate that *Rheb*$^{Y35L}$, *MTOR*$^{S2215Y}$, and *Tsc1*$^{KO}$ neurons display a decreased ability to generate APs upon depolarization despite having depolarized RMPs. In terms of firing pattern, neurons in all groups displayed a regular-spiking pattern with spike-frequency adaptation (*Figure 2e*). However, while an initial spike doublet was observed in control neurons, consistent with the expected firing pattern for layer 2/3 mPFC pyramidal neurons (*Kroon et al., 2019*), this was absent in all the experimental conditions except for the *Pten*$^{KO}$ condition (*Figure 2e*). Further quantification of the first interspike interval (first ISI; interval between the first and second AP) revealed significantly increased first ISI in *Rheb*$^{Y35L}$, *MTOR*$^{S2215Y}$, *Depdc5*$^{KO}$, and *Tsc1*$^{KO}$ neurons, but not in *Pten*$^{KO}$ neurons, compared to control neurons (*Figure 2h*, *Figure 2—figure supplement 1e*, *Figure 2—source data 1*). No differences in the AP threshold were observed across the conditions (*Figure 2i*, *Figure 2—figure supplement 1f*, *Figure 2—source data 1*). The AP peak amplitude was decreased in the *MTOR*$^{S2215Y}$ and *Tsc1*$^{KO}$ conditions (*Figure 2j*, *Figure 2—figure supplement 1g*, *Figure 2—source data 1*), while the AP half-width was decreased in the *Rheb*$^{Y35L}$, *MTOR*$^{S2215Y}$, and *Pten*$^{KO}$ conditions (*Figure 2k*, *Figure 2—figure supplement 1h*, *Figure 2—source data 1*). Taken together, these findings show that various genetic conditions that activate the mTORC1 pathway universally lead to decreased depolarization-induced excitability in layer 2/3 pyramidal neurons, with gene-dependent changes in RMP and several AP properties.

## Expression of *Rheb*$^{Y35L}$, *MTOR*$^{S2215Y}$, *Depdc5*$^{KO}$, *Pten*$^{KO}$, and *Tsc1*$^{KO}$ leads to the abnormal presence of HCN4 channels with variations in functional expression

We recently reported that neurons expressing *Rheb*$^{S16H}$, an mTORC1-activating variant of Rheb, display abnormal expression of HCN4 channels (*Hsieh et al., 2020*; *Nguyen et al., 2022*). These channels give rise to a hyperpolarization-activated cation current ($I_h$) that is normally absent in layer 2/3 pyramidal neurons (*Hsieh et al., 2020*; *Nguyen et al., 2022*). The aberrant $I_h$, which has implications for neuronal excitability, preceded seizure onset and was detected by P8-12 in mice (*Hsieh et al., 2020*). Rapamycin treatment starting at P1 and expression of constitutive active 4E-BP1, a translational repressor downstream of mTORC1, prevented and alleviated the aberrant HCN4 channel expression, respectively

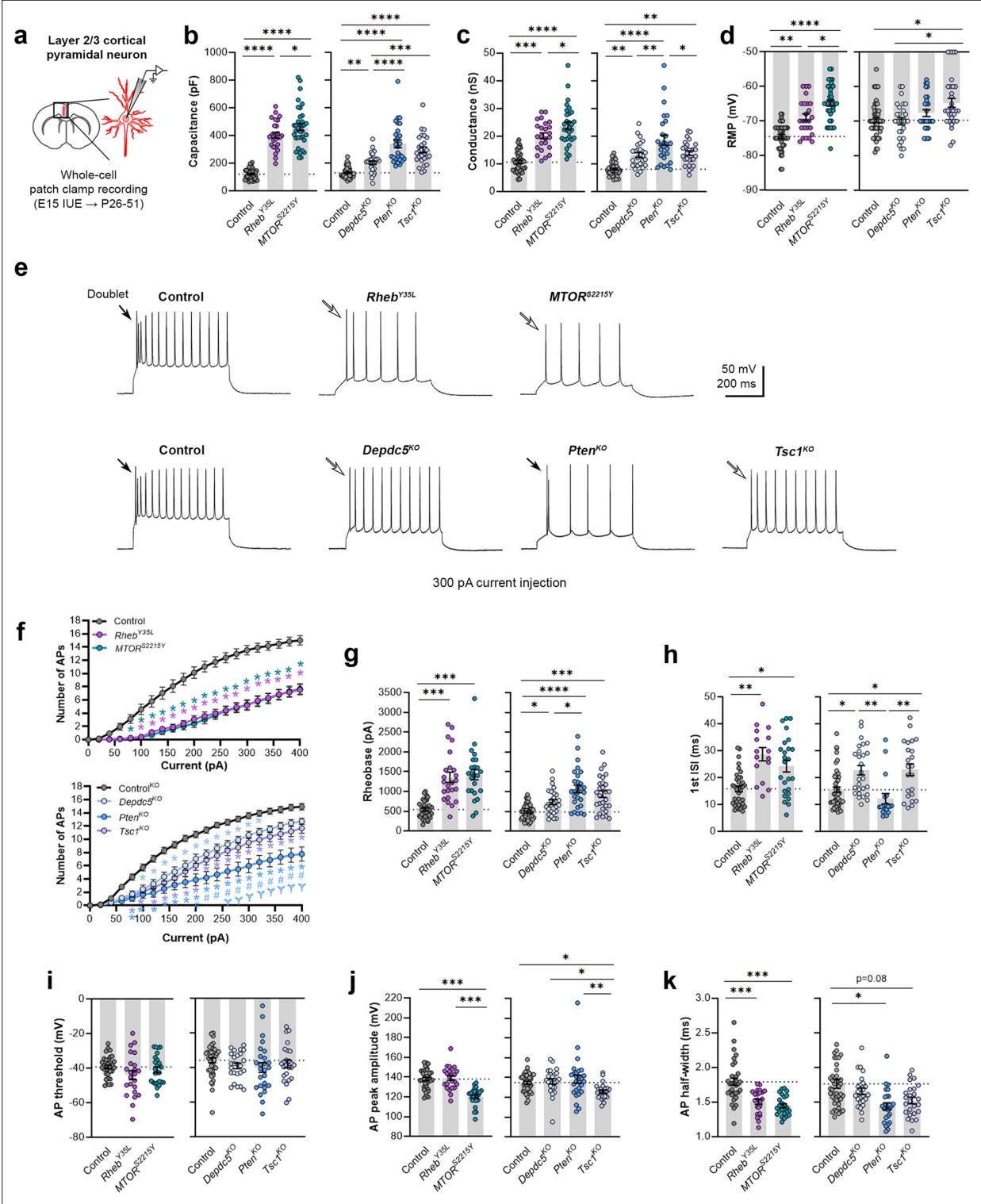

**Figure 2.** Expression of *Rheb^Y35L*, *MTOR^S2215Y*, *Depdc5^KO*, *Pten^KO*, and *Tsc1^KO* universally leads to decreased depolarization-induced excitability, but only *Rheb^Y35L*, *MTOR^S2215Y*, and *Tsc1^KO* expression leads to depolarized resting membrane potentials (RMPs). (**a**) Diagram of experimental approach for whole-cell patch clamp recording. Recordings were performed in layer 2/3 pyramidal neurons in acute coronal slices from P26-51 mice, expressing either control, *Rheb^Y35L*, *MTOR^S2215Y*, *Depdc5^KO*, *Pten^KO*, or *Tsc1^KO* plasmids. (**b–d**) Graphs of membrane capacitance, resting membrane conductance, and RMP. (**e**) Representative traces of the action potential (AP) firing response to a 300 pA depolarizing current injection. (**f**) Input-output curves show the mean number of APs fired in response to 500 ms-long depolarizing current steps from 0 to 400 pA. Arrows point to initial spike doublets. (**g–k**) Graphs of rheobase, first ISI, AP threshold, AP peak amplitude, and AP half-width. For all graphs: n=5–10 mice per group, with 16–50 cells analyzed per animal.

*Figure 2 continued on next page*

*Figure 2 continued*

Statistical comparisons were performed using (**b–d, g–k**) nested one-way ANOVA (fitted to a mixed-effects model) to account for correlated data within individual animals or (**f**) mixed-effects ANOVA accounting for repeated measures. Post-hoc analyses were performed using Holm-Šídák multiple comparison test. *p<0.05, **p<0.01, ***p<0.001, ****p<0.0001, for the input-output curves in (**f**): *p<0.05 (vs. control), #<0.05 (vs. *Depdc5*$^{KO}$), Ῡp<0.05 (vs. *Tsc1*$^{KO}$). All data are reported as the mean of all neurons in each group ± SEM.

The online version of this article includes the following source data and figure supplement(s) for figure 2:

**Source data 1.** Summary statistics for *Figure 2*.

**Figure supplement 1.** Distribution of membrane and action potential (AP) properties among individual animals at P26-51.

(*Hsieh et al., 2020*; *Nguyen et al., 2022*). These findings suggest that the abnormal HCN4 channel expression is mTORC1-dependent. Given that all the experimental conditions in the present study led to increased mTORC1 activity, we investigated whether they also resulted in abnormal HCN4 channel expression. Immunostaining for HCN4 channels using previously validated antibodies (*Hsieh et al., 2020*) in brain sections from P28-43 mice revealed significantly increased HCN4 staining intensity in the electroporated neurons in all experimental conditions compared to the respective controls, which exhibited no HCN4 staining (*Figure 3a and b*, *Figure 3—figure supplement 1a*, *Figure 3—source data 1*). The increased staining was evident in the ipsilateral cortex containing *MTOR*$^{S2215Y}$ electroporated neurons and absent in the non-electroporated contralateral cortex (*Figure 3—figure supplement 2a and b*). These data indicate the presence of aberrant HCN4 channel expression following *Rheb*$^{Y35L}$, *MTOR*$^{S2215Y}$, *Depdc5*$^{KO}$, *Pten*$^{KO}$, or *Tsc1*$^{KO}$ expression in layer 2/3 pyramidal neurons.

To examine the functional impacts of the aberrant HCN4 channel expression, we examined $I_h$ amplitudes in the various experimental conditions. To evoke $I_h$, we applied a series of 3 s-long hyperpolarizing voltage steps from –120 mV to –40 mV. Consistent with previous findings in *Rheb*$^{S16H}$ neurons (*Hsieh et al., 2020*; *Nguyen et al., 2022*), hyperpolarizing voltage pulses elicited significantly larger inward currents in all experimental conditions compared to their respective controls (*Figure 3c and d*, *Figure 3—source data 1*). The *MTOR*$^{S2215Y}$ condition displayed larger inward currents than the *Rheb*$^{Y35L}$ condition, while the *Pten*$^{KO}$ condition displayed the largest inward currents compared to the *Depdc5*$^{KO}$ and *Tsc1*$^{KO}$ conditions (*Figure 3d*). These data were proportional to the changes in neuron soma size (*Figures 1e and 2b*). The inward currents in *Rheb*$^{S16H}$ neurons are thought to result from the activation of both inward-rectifier K$^+$ (Kir) channels and HCN channels (*Hsieh et al., 2020*). Kir-mediated currents activate fast whereas HCN-mediated currents, i.e., $I_h$, activate slowly during hyperpolarizing steps; therefore, to assess $I_h$ amplitudes, we measured the difference between the instantaneous and steady-state currents at the beginning and end of the voltage pulses, respectively (i.e. ΔI) (*Thoby-Brisson et al., 2000*). The resulting ΔI-voltage (V) curve revealed significantly larger $I_h$ amplitudes in all experimental conditions (*Figure 3e*, *Figure 3—source data 1*). To further isolate the $I_h$ from Kir -mediated currents, we measured $I_h$ amplitudes at –90 mV, near the reversal potential of Kir channels to eliminate Kir-mediated current contamination. At –90 mV, $I_h$ amplitudes were significantly higher in the *MTOR*$^{S2215Y}$ and *Tsc1*$^{KO}$ conditions compared to controls (*Figure 3f*, *Figure 3—figure supplement 1b*, *Figure 3—source data 1*). Of note, although the *Depdc5*$^{KO}$ and *Pten*$^{KO}$ conditions did not display a significant increase in $I_h$ amplitudes at –90 mV, 1 out of 28 *Depdc5*$^{KO}$ neurons and 4 out of 27 *Pten*$^{KO}$ neurons had $I_h$ amplitudes that were twofold greater than the mean $I_h$ amplitude of the *Tsc1*$^{KO}$ condition. 6 out of 24 *Rheb*$^{Y35L}$ neurons also had $I_h$ amplitudes twofold greater than this value (*Figure 3f*). These data suggest that $I_h$ currents are present in a subset of *Rheb*$^{Y35L}$, *Depdc5*$^{KO}$, and *Pten*$^{KO}$ neurons, and most *MTOR*$^{S2215Y}$ and *Tsc1*$^{KO}$ neurons.

Considering that the *MTOR*$^{S2215Y}$ condition led to the largest $I_h$, we examined the impact of the selective HCN channel blocker zatebradine on hyperpolarization-induced inward currents in *MTOR*$^{S2215Y}$ neurons to further confirm the identity of ΔI as $I_h$. Application of 40 μM zatebradine reduced the overall inward currents (*Figure 3g and h*, *Figure 3—source data 1*) and ΔI (*Figure 3i*, *Figure 3—source data 1*). At –90 mV, ΔI was significantly decreased from –167.7 ± 54.2 pA to 0.75 ± 8.2 pA (± SD) (*Figure 3i*, arrow). Subtraction of the post- from the pre-zatebradine current traces isolated zatebradine-sensitive inward currents which reversed near –50 mV, as previously reported for HCN4 channel reversal potentials (*Tae et al., 2017*; *Figure 3j and k*). These experiments verified the identity of ΔI as $I_h$. In comparison, application of the Kir channel blocker barium chloride (BaCl$_2$) substantially reduced the overall inward currents but had no effects on ΔI (i.e. $I_h$) in *Tsc1*$^{KO}$ neurons (*Figure 3—figure supplement 3a–i*). Consistent with the function of $I_h$ in maintaining RMP

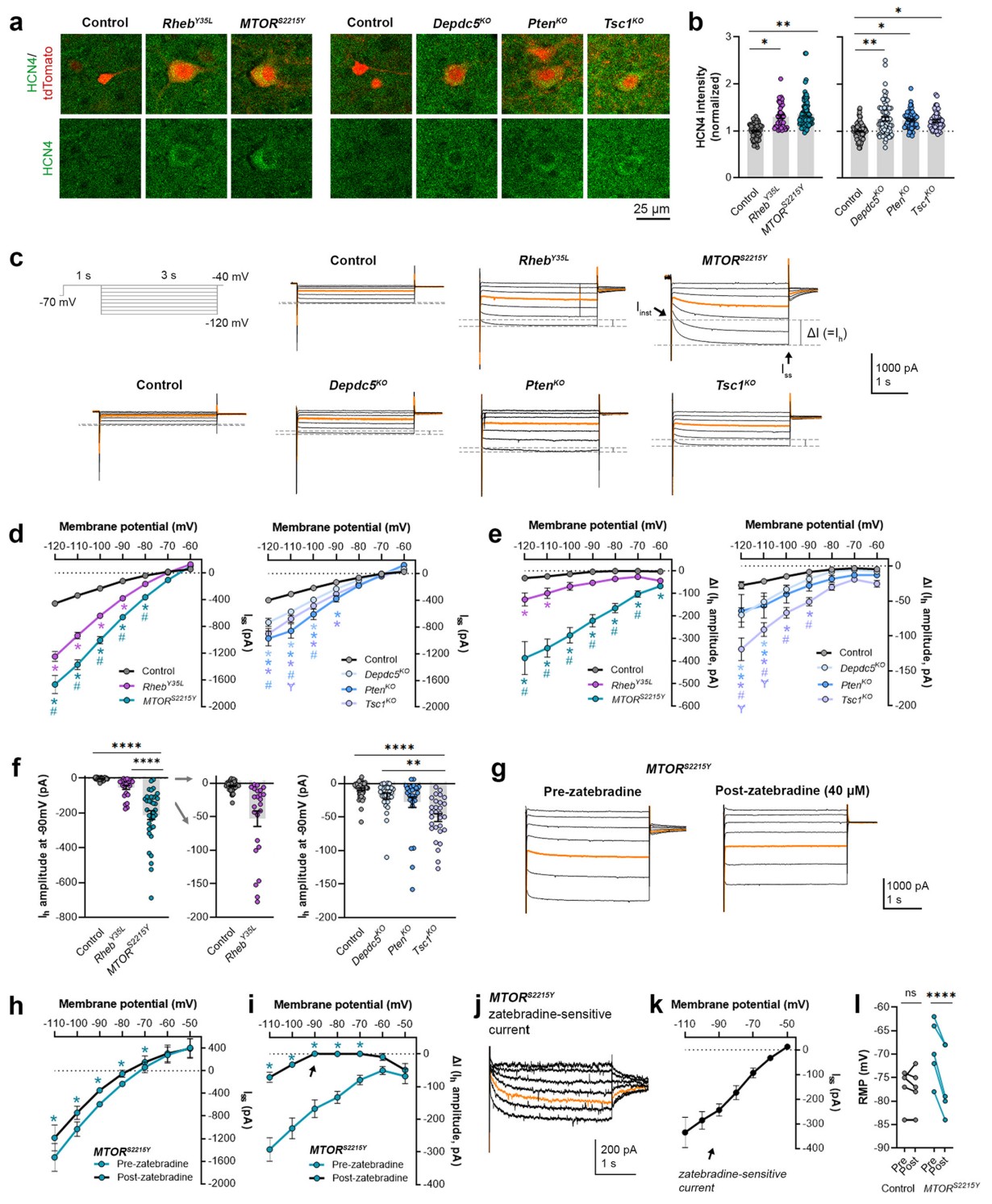

**Figure 3.** Expression of $Rheb^{Y35L}$, $MTOR^{S2215Y}$, $Depdc5^{KO}$, $Pten^{KO}$, and $Tsc1^{KO}$ leads to the abnormal presence of HCN4 channels with variations in functional expression. (**a**) Representative images of tdTomato+ cells (red) and HCN4 staining (green) in mouse medial prefrontal cortex (mPFC) at P28-43. (**b**) Quantification of HCN4 intensity (normalized to the mean control) in tdTomato+ neurons. (**c**) Representative current traces in response to a series of 3 s-long hyperpolarizing voltage steps from –120 to –40 mV, with a holding potential of –70 mV. Current traces from the –40 and –50 mV steps were not included due to contamination from unclamped Na+ spikes. Orange lines denote the current traces at –90 mV. (**d**) IV curve obtained from $I_{ss}$ amplitudes. (**e**) ΔIV curve obtained from $I_h$ amplitudes (i.e. ΔI, where $ΔI=I_{ss} – I_{inst}$). (**f**) Graphs of $I_h$ amplitudes at –90 mV. (**g**) Representative current traces in response to a series of 3 s long hyperpolarizing voltage steps from –110 mV to –50 mV in the $MTOR^{S2215Y}$ condition pre- and post-zatebradine

*Figure 3 continued on next page*

*Figure 3 continued*

application. Orange lines denote the current traces at –90 mV. (**h**) IV curve obtained from $I_{ss}$ amplitudes in the *MTOR^S2215Y^* condition pre- and post-zatebradine application. (**i**) ΔIV curve obtained from $I_h$ amplitudes (i.e. ΔI) in the *MTOR^S2215Y^* condition pre- and post-zatebradine application. Arrow points to the post-zatebradine $I_h$ amplitude at –90 mV. (**j**) Representative traces of the zatebradine-sensitive current obtained after subtraction of the post- from the pre-zatebradine current traces in response to –110 mV to –50 mV voltage steps. Orange lines denote the current traces at –90 mV. (**k**) IV curve of the zatebradine-sensitive current obtained after subtraction of the post- from the pre-zatebradine current traces. (**l**) Graph of RMP in the control and *MTOR^S2215Y^* conditions pre- and post-zatebradine application. Connecting lines denote paired values from the same neuron. For graph (**b**): n=4–8 mice per group, with 4–15 cells analyzed per animal. For graphs **d, e, f**: n=5–10 mice per group, with 24–47 cells analyzed per animal. For graphs **h, i, k, l**: n=4–6 neurons (paired). Statistical comparisons were performed using (**b, f**) nested ANOVA (fitted to a mixed-effects model) to account for correlated data within individual animals, (**d, e**) mixed-effects ANOVA accounting for repeated measures, or (**h, i, l**) two-way repeated measures ANOVA. Post-hoc analyses were performed using Holm-Šídák multiple comparison test. *p<0.05, **p<0.01, ****p<0.0001, for the IV curves in **d, e, h, i**: *p<0.05 (vs. control), #p<0.05 (vs. *Rheb^Y35L^* or *Depdc5^KO^*), ϒp<0.05 (vs. *Pten^KO^*). All data are reported as the mean of all neurons in each group ± SEM.

The online version of this article includes the following source data and figure supplement(s) for figure 3:

**Source data 1.** Summary statistics for *Figure 3*.

**Figure supplement 1.** Distribution of HCN4 staining intensity and hyperpolarization-activated cation current ($I_h$) amplitudes (at –90 mV) among individual animals at P28-43 and P26-51, respectively.

**Figure supplement 2.** Additional images of HCN4 staining in medial prefrontal cortex (mPFC) sections from P28-43 mice expressing *MTOR^S2215Y^*.

**Figure supplement 3.** BaCl₂ application decreases overall inward currents without affecting hyperpolarization-activated cation current ($I_h$) in *Tsc1^KO^* neurons.

**Figure supplement 3—source data 1.** Summary statistics for *Figure 3—figure supplement 3*.

at depolarized levels (*Lamas, 1998*; *Doan and Kunze, 1999*; *Lupica et al., 2001*; *Funahashi et al., 2003*; *Kase and Imoto, 2012*), the application of zatebradine hyperpolarized RMP in *MTOR^S2215Y^* neurons but did not affect the RMP of control neurons that exhibited no $I_h$ (*Figure 3l*, *Figure 3— source data 1*). Collectively, these findings suggest that *Rheb^Y35L^*, *MTOR^S2215Y^*, *Depdc5^KO^*, *Pten^KO^*, and *Tsc1^KO^* expression in layer 2/3 pyramidal neurons lead to the abnormal presence of HCN4 channels with variations in functional expression.

## Expression of *Rheb^Y35L^*, *MTOR^S2215Y^*, *Depdc5^KO^*, *Pten^KO^*, and *Tsc1^KO^* leads to different impacts on excitatory synaptic activity

As part of examining neuron excitability, we recorded spontaneous excitatory postsynaptic currents (sEPSCs) in all the gene conditions. To separate sEPSCs from spontaneous inhibitory postsynaptic currents (sIPSCs), we used an intracellular solution rich in K-gluconate to impose a low intracellular Cl⁻ concentration and recorded at a holding potential of –70 mV, which is near the Cl⁻ reversal potential. The 90–10% decay time of the measured synaptic currents ranged between 4–8 ms in all conditions (mean ± SD: control: 4.9 ± 1.6; *Rheb^Y35L^*: 5.2 ± 1.4; *MTOR^S2215Y^*: 7.4 ± 1.4; control: 6.8 ± 0.7; *Depdc5^KO^*: 7.4 ± 1.0; *Pten^KO^*: 8.1 ± 0.9; *Tsc1^KO^*: 7.4 ± 0.9), consistent with the expected decay time for sEPSCs and shorter than the decay time for sIPSCs (*Kroon et al., 2019*). The recorded synaptic currents were, therefore, considered to be sEPSCs. The sEPSCs frequency was unchanged in all experimental conditions except for the *Tsc1^KO^* condition, where the sEPSCs frequency was significantly increased (*Figure 4a and b*, *Figure 4—figure supplement 1a*, *Figure 4—source data 1*). Unlike the other experimental conditions, the *Rheb^Y35L^* condition displayed a slight decrease in sEPSC frequency, consistent with previous findings in *Rheb^S16H^* neurons; however, this did not reach statistical significance (*Figure 4b*). The sEPSC amplitude was larger in the *Rheb^Y35L^*, *MTOR^S2215Y^*, and *Pten^KO^* conditions (*Figure 4a and c*, *Figure 4—figure supplement 1b*, *Figure 4—source data 1*). Although the amplitudes were slightly larger in the *Depdc5^KO^* and *Tsc1^KO^* conditions, these changes were not significant (*Figure 4c*). Thus, *Tsc1^KO^* neurons display increased sEPSC frequency but unchanged amplitude, while *Rheb^Y35L^*, *MTOR^S2215Y^*, and *Pten^KO^* neurons display increased sEPSC amplitude but unchanged frequency. Finally, there was an increase in the sEPSC total charge in all experimental conditions except for the *Rheb^Y35L^* condition (*Figure 4d*, *Figure 4—figure supplement 1c*, *Figure 4—source data 1*). Collectively, these findings suggest all experimental conditions, except for *Rheb^Y35L^*, lead to increased synaptic excitability, with variable impact on sEPSC frequency and amplitude.

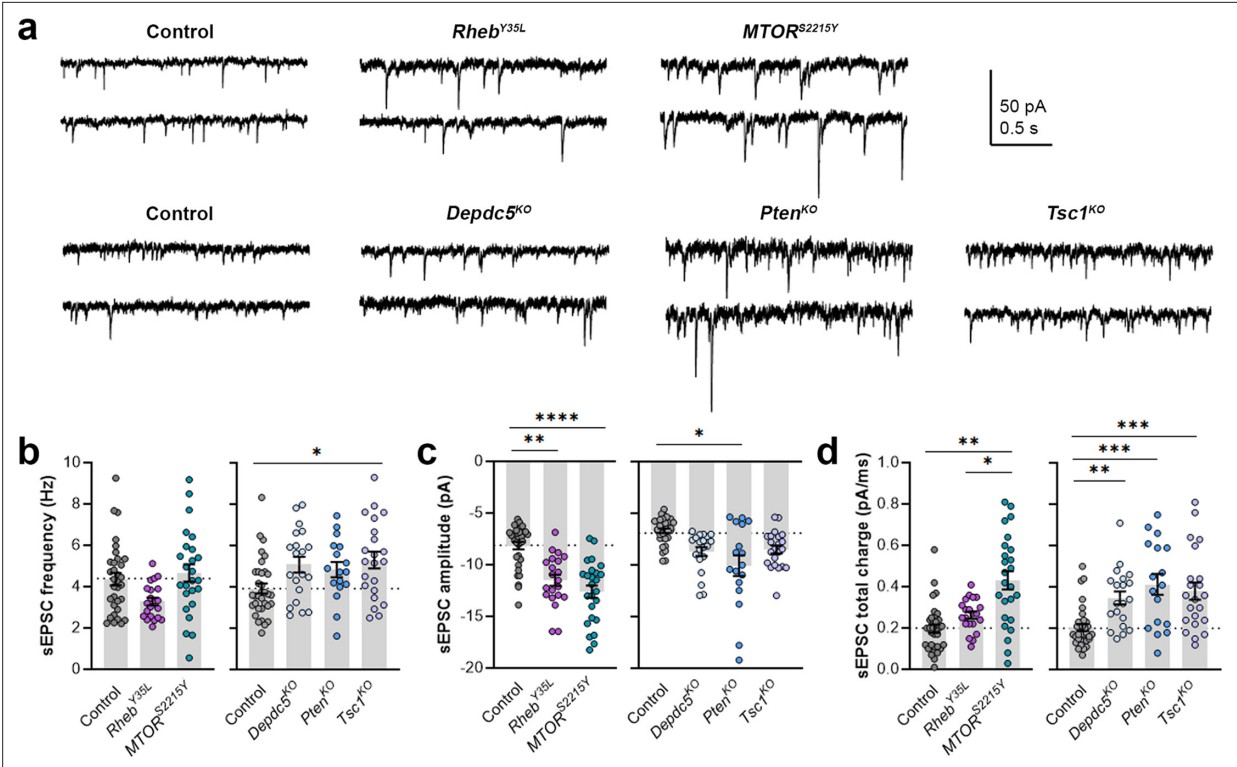

**Figure 4.** Expression of *Rheb*$^{Y35L}$, *MTOR*$^{S2215Y}$, *Depdc5*$^{KO}$, *Pten*$^{KO}$, and *Tsc1*$^{KO}$ leads to different impacts on spontaneous excitatory postsynaptic current (sEPSC) properties. (**a**) Representative sEPSC traces were recorded at a holding voltage of –70 mV. Top and bottom traces are from the same neuron. (**b–d**) Graphs of sEPSC frequency, amplitude, and total charge. For all graphs: n=5–9 mice per group, with 17–34 cells analyzed per animal. Statistical comparisons were performed using a nested ANOVA (fitted to a mixed effects model) to account for correlated data within individual animals. Post-hoc analyses were performed using Holm-Šídák multiple comparison test. *p<0.05, **p<0.01, ***p<0.001, ****p<0.0001. All data are reported as the mean of all neurons in each group ± SEM.

The online version of this article includes the following source data and figure supplement(s) for figure 4:

**Source data 1.** Summary statistics for *Figure 4*.

**Figure supplement 1.** Distribution of spontaneous excitatory postsynaptic current (sEPSC) properties among individual animals at P26-51.

## Discussion

In this unprecedented comparison study, we examined the impacts of several distinct epilepsy-associated mTORC1 pathway gene mutations on cortical pyramidal neuron development and their electrophysiological properties and synaptic integration in the mouse mPFC. Through a combination of IUE to model the genetic mosaicism of FMCDs, histological analyses, and patch-clamp electrophysiological recordings, we found that activation of either *Rheb* or *MTOR* or biallelic inactivation *Depdc5*, *Tsc1*, or *Pten*, all of which increase mTORC1 activity, largely leads to similar alterations in neuron morphology and membrane excitability but differentially impacts excitatory synaptic transmission. These findings have significant implications for understanding gene-specific mechanisms leading to cortical hyperexcitability and seizures in mTORC1-driven FMCDs and highlight the utility of personalized medicine dictated by patient gene variants. Furthermore, our study emphasizes the importance of considering gene-specific approaches when modeling genetically distinct mTORopathies for basic research studies.

Several histological phenotypes associated with mTORC1 hyperactivation were anticipated and confirmed in our studies. We found increased neuron soma size across all gene conditions, consistent with previous reports (*Zhao et al., 2019*; *Ribierre et al., 2018*; *Lim et al., 2017*; *Chen et al., 2015*; *Lim et al., 2015*; *Hsieh et al., 2016*; *Proietti Onori et al., 2021*). Additionally, all conditions, except for *Pten*$^{KO}$, resulted in neuronal mispositioning in the mPFC, with *Rheb*$^{Y35L}$ and *MTOR*$^{S2215Y}$ conditions being the most severe. Interestingly, *Pten*$^{KO}$ neurons were correctly placed in layer 2/3, despite having one of the largest soma size increases, suggesting that cell positioning is independent of cell size.

The lack of mispositioning of $Pten^{KO}$ neurons was surprising as it is thought that increased mTORC1 activity leads to neuronal misplacement, and aberrant migration of $Pten^{KO}$ neurons has been reported in the hippocampus (*Getz et al., 2016*). Given that PTEN has a long half-life, with a reported range of 5–20 hr or more depending on the cell type (*Trotman et al., 2007*; *Mukherjee et al., 2021*; *Li et al., 2018*; *Wong et al., 2018*; *Yang et al., 2009*; *Yim et al., 2009*; *Vazquez et al., 2000*; *Sacco et al., 2014*), it is possible that following knockout at E15, the decreases in existing protein levels lagged behind the time window to affect neuronal migration. However, TSC1 also has a long half-life of ~22 hr (*Lee et al., 2008*; *Alquezar et al., 2021*), and knockout of *Tsc1* at the same time point was sufficient to affect neuronal positioning. Mispositioning of neurons following *Pten* knockout in E15 rats was reported by one study, but this was a very mild phenotype compared to findings in the other gene conditions (*Chen et al., 2015*). Therefore, these data suggest that PTEN has gene-specific biological mechanisms that contribute to the lack of severe neuronal mispositioning in the developing cortex. $MTOR^{S2215Y}$ exhibited the strongest phenotypes in terms of neuron size and positioning, which is perhaps not surprising as the mTOR protein itself forms the catalytic subunit of mTORC1. $Depdc5^{KO}$ exhibited the mildest changes in these parameters. Unlike PTEN, TSC1, RHEB, and mTOR, which regulate mTORC1 via the canonical PI3K-mTORC1 pathway in response to growth factor stimulation, DEPDC5 regulates mTORC1 via the GATOR1 complex in response to changes in cellular amino acid levels (*Iffland et al., 2019*). The different modes of mTORC1 regulation by DEPDC5 may contribute to the differences in the severity of the phenotypes.

To assess the impact of RHEB, MTOR, DEPDC5, PTEN, and TSC1 disruption on cortical neuron excitability, we examined the intrinsic biophysical and synaptic properties of layer 2/3 pyramidal neurons in the mPFC. We found that all gene manipulations led to increased membrane conductance but decreased AP firing upon membrane depolarization, consistent with previous reports in the somatosensory cortex (*Ribierre et al., 2018*; *Chen et al., 2015*; *Proietti Onori et al., 2021*; *Hu et al., 2018*; *Goz et al., 2020*; *Koh et al., 2021*; *Wu et al., 2022*) and in the mPFC for the $Rheb^{S16H}$ condition (*Hsieh et al., 2020*; *Nguyen et al., 2022*). Neurons in all experimental conditions required a larger depolarization to generate an AP (increased rheobase), likely due to their enlarged cell size. These findings seem to contradict the theory that seizure initiation originates from these neurons. However, inhibiting firing in cortical pyramidal neurons expressing the $Rheb^{S16H}$ mutation via co-expression of Kir2.1 channels (to hyperpolarize neurons and shunt their firing) has been shown to prevent seizures, supporting a cell-autonomous mechanism (*Hsieh et al., 2020*). This discrepancy may be reconciled by the identification of abnormal HCN4 channel expression in these neurons, which contributed to increased excitability by depolarizing RMPs, i.e., bringing the neurons closer to the AP threshold, and conferring firing sensitivity to intracellular cyclic AMP (*Hsieh et al., 2020*). The abnormal expression of HCN4 channels in the $Rheb^{S16H}$ neurons was found to be mTORC1-dependent by their sensitivity to rapamycin and the expression of constitutive active 4E-BP1. In addition, the abnormal HCN4 channel expression has been verified in resected human FCDII and HME tissue (of unknown genetic etiology) (*Hsieh et al., 2020*; *Nguyen et al., 2022*). Here, we found that $Rheb^{Y35L}$, $MTOR^{S2215Y}$, $Depdc5^{KO}$, $Pten^{KO}$, and $Tsc1^{KO}$ cortical neurons also display aberrant HCN4 channel expression, which is consistent with the mTORC1-dependent mode of expression in $Rheb^{S16H}$ neurons (*Hsieh et al., 2020*; *Nguyen et al., 2022*). The functional expression of these channels was variable, and the corresponding changes in $I_h$ did not reach statistical significance for $Rheb^{Y35L}$, $Depdc5^{KO}$, and $Pten^{KO}$ neurons. Nonetheless, consistent with the function of HCN4 channels in maintaining the RMP at depolarized levels (*Kase and Imoto, 2012*), $Rheb^{Y35L}$, $MTOR^{S2215Y}$, and $Tsc1^{KO}$ neurons exhibited depolarized RMPs. Interestingly, the RMP was unchanged in $Pten^{KO}$ and $Depdc5^{KO}$ neurons. The lack of RMP changes may be explained by the milder HCN4 phenotype or the presence of different ion channel complements in these neurons. In addition to HCN4 channels, it is thought that increased Kir channels, which are well-known to hyperpolarize RMPs, contribute to the enlarged inward currents in all the conditions we investigated. This was confirmed in the $Tsc1^{KO}$ neurons where the application of $BaCl_2$ to block Kir channels reduced the overall inward current and depolarized RMPs. Since $Pten^{KO}$ neurons have a larger overall inward current but a smaller $I_h$ compared to $Tsc1^{KO}$ neurons, we postulate that $Pten^{KO}$ neurons have a higher expression of Kir channels neurons that could counteract the HCN4 channels' depolarizing effect on RMP. The mechanisms and functional significance of the Kir channel increases in these neurons are yet to be validated. Furthermore, studies using rapamycin in each of the gene conditions would provide more direct evidence for the mTORC1-dependency of the HCN4 channel expression. Nevertheless,

our data suggests that abnormal HCN4 channel expression is a conserved mechanism across these mTORC1-activating gene conditions and warrants further investigation of HCN-mediated excitability in mTORC1-related epilepsy.

While membrane excitability was largely similar across all gene conditions, the excitatory synaptic properties were more variable. All conditions, except for $Rheb^{Y35L}$, led to increased excitatory synaptic activity, with variable impact on sEPSC frequency and amplitude. $Tsc1^{KO}$ neurons were the only investigated condition that displayed increased sEPSC frequency. This finding corroborates a previous study showing increased sEPSC frequency with no sEPSC amplitude changes in layer 2/3 cortical pyramidal neurons from 4-week-old $Tsc1$ conditional KO mice (**Wu et al., 2022**). Interestingly, as previously reported for $Rheb^{S16H}$ neurons (**Lin et al., 2016**), the sEPSC frequency in $Rheb^{Y35L}$ neurons was reduced by 25% but this did not reach statistical significance, possibly due to a generally weaker effect of the Y35L mutation compared to the S16H mutation. $Rheb^{Y35L}$, $MTOR^{S2215Y}$, and $Pten^{KO}$ neurons exhibited increased sEPSC amplitudes. We found no changes in the sEPSC amplitude of $Depdc5^{KO}$ neurons, which differs from a previous study reporting increased sEPSC amplitude in these neurons (**Ribierre et al., 2018**). This discrepancy may likely be explained by the differences in the age of the animals examined in their study (P20-24) versus ours (P26-43), given that dynamic changes in synaptic properties are still ongoing past P21 (**Lohmann and Kessels, 2014**). Despite the above differences in sEPSC frequencies and amplitudes, the total charge was increased in almost all conditions except for in $Rheb^{Y35L}$, suggesting enhanced synaptic drive and connectivity. These excitatory synaptic changes may counteract the increases in rheobase (i.e. decreased membrane excitability) and thereby impact circuit hyperexcitability and seizure vulnerability. Overall, the changes in synaptic activity for the $Rheb^{Y35L}$ condition are in stark contrast to the other gene conditions. These observations suggest a Rheb-specific impact on synaptic activity that differs from the other mTORC1 pathway genes. Thus, despite impinging on mTORC1 signaling, different mTORC1 pathway gene mutations can affect synaptic activity, and thereby, excitability differently. The mechanisms accounting for the observed differences are not clear, but it is increasingly acknowledged that additional pathways beyond mTORC1 are activated or inactivated by each gene condition that could potentially contribute to the differential impacts on synaptic activity (**Nguyen and Bordey, 2021**).

Although all the examined gene conditions activate mTORC1 signaling, they differentially impact the mTORC2 pathway. mTORC2 is a lesser understood, acute rapamycin-insensitive complex formed by MTOR binding to Rictor and known to regulate distinct cellular functions, including actin cytoskeletal organization (**Laplante and Sabatini, 2012**). Loss of PTEN increases mTORC2 activity, whereas loss of TSC1/2 and DEPDC5 is associated with decreased mTORC2 activity (**Zhou et al., 2009**; **Nguyen et al., 2015**; **Huang et al., 2008**; **Yuskaitis et al., 2018**; **Karalis et al., 2022**). $MTOR^{S2215Y}$ and $Rheb^{Y35L}$ are not expected to change mTORC2 activity, although this remains to be verified. Studies have shown that inhibition of mTORC2 reduces neuronal overgrowth but not the synaptic defects and seizures associated with PTEN loss (**Cullen et al., 2023**; **Cullen et al., 2024**). However, other studies have reported that inhibition of mTORC2 reduces seizures in several epilepsy models, including the $MTOR^{S2215F}$ gain-of-function and $Pten$ KO models (**Chen et al., 2019**; **Okoh et al., 2023**). Thus, the contribution of mTORC2 in epileptogenesis and seizure generation remains unclear, and future studies aimed to address the contribution of mTORC2 to the neuronal properties and synaptic activity in the different gene conditions are important to pursue.

Electrophysiological data from cytomegalic pyramidal neurons in cortical tissue obtained from humans with FCDII or TSC and intractable seizures have been reported (**Mathern et al., 2000**; **Cepeda et al., 2003**; **Cepeda et al., 2005**; **Cepeda et al., 2010**). These studies showed that human cytomegalic neurons have increased capacitance and decreased input resistance (**Cepeda et al., 2003**; **Cepeda et al., 2005**), consistent with our findings in the present and previous studies in mice (**Hsieh et al., 2020**; **Nguyen et al., 2022**). Further comparisons between pyramidal neurons in human TSC and FCDII cases showed similar changes in passive membrane and firing properties but differences in sEPSCs properties between the two conditions (**Cepeda et al., 2010**). In particular, the frequency of sEPSC was higher in neurons in TSC compared to FCDII. These data were similar to our findings showing increased sEPSC frequency in the $Tsc1^{KO}$ condition but not in the other key gene conditions associated with FCDII. The authors of the 2010 human electrophysiology study concluded that although TSC and FCDII share several histopathologic similarities, there are subtle functional differences between these disorders (**Cepeda et al., 2010**), aligning with the overall conclusions

from our study. However, because the genetic etiology of the FCDII cases in these human studies is unknown, it is not possible to fully compare our gene-specific data to information published in these studies. Another study has reported a functional reduction of $GABA_A$-mediated synaptic transmission in cortical pyramidal neurons in individuals with FCDII (**Calcagnotto et al., 2005**). In our study, we did not examine inhibitory synaptic properties and the impact on excitatory-inhibitory balance; this would be an important subject to pursue in another study.

Brain somatic mutations causing FMCDs, such as FCDII and HME, occur throughout cortical neurogenesis in neural progenitor cells that give rise to excitatory (pyramidal) neurons (**D'Gama et al., 2017**; **Chung et al., 2023**). In the present study, we performed electroporation at E15, which targets progenitor cells that generate layer 2/3 pyramidal neurons in mice. Given that somatic mutations in FMCDs may occur at various timepoints during brain development, it would be interesting to examine the effects of mTOR pathway mutations in other cortical neuronal populations. For example, it would be interesting to investigate whether targeting earlier-born, layer 5 neurons by electroporating at E13 would result in similar or distinct phenotypes compared to our present observations in layer 2/3 neurons, and whether this would mitigate or accentuate the differences between the gene conditions. These findings would provide further insights into somatic mutations and mechanisms of FMCD and epilepsy.

In summary, we have shown that mutations affecting different mTORC1 pathway genes have similar and dissimilar consequences on cortical pyramidal neuron development and function, which may affect how neurons behave in cortical circuits. Our findings suggest that cortical neurons harboring different mTORC1 pathway gene mutations may differentially affect how neurons receive and process cortical inputs, which have implications for the mechanisms of cortical hyperexcitability and seizures in FMCDs, and potentially affect how neurons and their networks respond to therapeutic intervention.

## Materials and methods

**Key resources table**

| Reagent type (species) or resource | Designation | Source or reference | Identifiers | Additional information |
|---|---|---|---|---|
| Strain, strain background (*Mus musculus*) | CD1, wildtype (time-pregnant females) | Charles River Laboratories | Strain code: 022, RRID:IMSR_CRL:022 | For in utero electroporation |
| Recombinant DNA reagent | pCAG-tdTomato (plasmid) | Addgene | Plasmid #83029, RRID:Addgene_83029 | Deposited by Dr. Angelique Bordey (previously generated by our lab) |
| Recombinant DNA reagent | pCAG-GFP (plasmid) | Addgene | Plasmid #11150, RRID:Addgene_11150 | Deposited by Dr. Connie Cepko |
| Recombinant DNA reagent | pCAG-tagBFP (plasmid) | Dr. Joshua Breunig, PMID:22776033 | - | Gift from Dr. Joshua Breunig. The generation of this plasmid has previously been described (**Breunig et al., 2012**). |
| Recombinant DNA reagent | pCAG-*Rheb*$^{Y35L}$-T2A-tdTomato (plasmid) | This paper | - | The *Rheb* insert was synthesized and subcloned into a pCAG-Kir2.1Mut-T2A-tdTomato construct (Addgene plasmid #60644) via SmaI and BamHI sites, after excision of Kir2.1. ***Rheb* insert sequence:** NM_053075.3 (*Mus musculus*), mutation: Y35L |
| Recombinant DNA reagent | pCAG-*MTOR*$^{S2215Y}$ (plasmid) | This paper | - | The *MTOR* insert was subcloned from a pcDNA3-FLAG-mTOR-SS2215Y construct (Addgene plasmid #69013) into a pCAG-tdTomato construct (Addgene plasmid #83029) via 5'AgeI and 3'NotI sites, after excision of tdTomato. ***MTOR* insert sequence:** NM_004958 (*Homo sapiens*), mutation: S2215Y |
| Recombinant DNA reagent | pX330-Control (pX330-empty) (plasmid) | Addgene | Plasmid #42230, RRID:Addgene_42230 | Deposited by Dr. Feng Zhang |

*Continued on next page*

*Continued*

| Reagent type (species) or resource | Designation | Source or reference | Identifiers | Additional information |
|---|---|---|---|---|
| Recombinant DNA reagent | pX330-*Depdc5* (plasmid) | Dr. Stéphanie Baulac, PMID:29708508 | - | Gift from Dr. Stéphanie Baulac. The gRNA sequence (gRNA1) has previously been validated (*Ribierre et al., 2018*). gRNA sequence (5′ to 3′): GTCTGTGGTGATCACGC (17 nt) The gRNA targets exon 16 of *Depdc5* (NM_001025426, *Mus musculus*), resulting in deleterious indels and inactivation of the gene |
| Recombinant DNA reagent | pX330-*Pten* (plasmid) | Addgene | Plasmid #59909, RRID:Addgene_59909 | Deposited by Dr. Tyler Jacks. The gRNA sequence (gRNA1) has previously been validated (*Xue et al., 2014*). gRNA sequence (5′ to 3′): AGATCGTTAGCAGAAACAAA (20 nt) The gRNA targets *Pten* (NM_008960, *Mus musculus*), resulting in deleterious indels and inactivation of the gene |
| Recombinant DNA reagent | pX330-*Tsc1* (plasmid) | This paper | - | The gRNA sequence was synthesized based on previously published sequences (*Lim et al., 2017*) and subcloned into a pX330-empty vector. The specificity of the gRNA (gRNA T4) has previously been described and validated (*Lim et al., 2017*). gRNA sequence (5′ to 3′): CAGTGTGGAGGAGTCCAGCA (20 nt) The gRNA targets exon 3 of *Tsc1* (NM_001289575, *Mus musculus*), resulting in deleterious indels and inactivation of the gene |
| Antibody | anti-p-S6 S240/244 (rabbit monoclonal) | Cell Signaling Technology | Cat # 5364, RRID:AB_10694233 | IF (1:1000) |
| Antibody | anti-HCN4 (rabbit polyclonal) | Alomone Labs | Cat # APC-052, RRID:AB_2039906 | IF (1:500) |
| Antibody | goat anti-rabbit IgG, Alexa Fluor Plus 647 | Invitrogen | Cat # A32733, RRID:AB_2633282 | IF (1:500) |
| Software, algorithm | Image J | NIH | RRID:SCR_003070 | - |
| Software, algorithm | Photoshop | Adobe | RRID:SCR_014199 | Version CC |
| Software, algorithm | pClamp | Molecular Devices | RRID:SCR_011323 | Version 10 |
| Software, algorithm | Prism | GraphPad Software | RRID:SCR_002798 | Version 9 |

## Animals

All animal procedures were performed in accordance with Yale University Institutional Animal Care and Use Committee's regulations and approved protocols (protocol number: 2022–10514). In utero electroporation was performed on time-pregnant, female CD-1 mice (Charles River Laboratories, strain code: 022). All subsequent experiments were performed on male and female offspring of the electroporated dams.

## Plasmid DNA

Information on the plasmids used in this study is listed in the Key resources table. The concentration of the plasmids used for in utero electroporation ranged from 0.5 to 3.5 µg/µl. The specific concentrations and combinations of plasmids used for each of the control and experimental conditions are listed in *Table 1*.

## In utero electroporation (IUE)

Timed-pregnant embryonic day (E) 15.5 mice were anesthetized with isoflurane, and a midline laparotomy was performed to expose the uterine horns. A DNA plasmid solution (1.5 µl) was injected into the right lateral ventricle of each embryo using a glass pipette. For each litter, half of the embryos received plasmids for the experimental condition and the other half received plasmids for the

**Table 1.** Plasmid concentrations for control and experimental conditions.

| Group | Plasmid name | Plasmid concentration (μg/μl)* |
|---|---|---|
| Control (for $Rheb^{Y35L}$ littermates)† | pCAG-tdTomato | 3.5 |
| Control (for $MTOR^{S2215Y}$ littermates)† | pCAG-GFP | 2.5 |
| | pCAG-tdTomato | 1.0 |
| | pCAG-$Rheb^{Y35L}$-T2A-tdTomato | 2.5 |
| $Rheb^{Y35L}$ | pCAG-GFP | 1.0 |
| | pCAG-$MTOR^{S2215Y}$ | 2.5 |
| $MTOR^{S2215Y}$ | pCAG-tdTomato | 1.0 |
| | pX330-Control | 2.5 |
| | pCAG-tdTomato | 1.0 |
| Control | pCAG-tagBFP‡ | 0.5 |
| | pX330-$Depdc5$ | 2.5 |
| | pCAG-tdTomato | 1.0 |
| $Depdc5^{KO}$ | pCAG-GFP‡ | 0.5 |
| | pX330-$Pten$ | 2.5 |
| | pCAG-tdTomato | 1.0 |
| $Pten^{KO}$ | pCAG-GFP‡ | 0.5 |
| | pX330-$Tsc1$ | 2.5 |
| | pCAG-tdTomato | 1.0 |
| $Tsc1^{KO}$ | pCAG-GFP‡ | 0.5 |

*Working plasmid solutions were diluted in water and contained 0.03% Fast Green dye to visualize the injection. For each embryo, 1.5 μl of the plasmid mixture was injected into the right ventricle.

†Control groups for $Rheb^{Y35L}$ and $MTOR^{S2215Y}$ contained pCAG tdTomato and pCAG-GFP +pCAG-tdTomato, respectively, to distinguish between control and experimental animals for each litter. The total plasmid concentration was kept equal between the two control groups. No differences were observed between the two groups, and therefore, they were combined into one control group.

‡Equal concentrations of pCAG-tagBFP and pCAG-GFP were added into the control and experimental ($Depdc5^{KO}$, $Pten^{KO}$, and $Tsc1^{KO}$) groups, respectively, to distinguish the control and experimental animals for each litter.

respective control condition. A 5 mm tweezer electrode was positioned on the embryo head and 6 × 42 V, 50 ms pulses at 950 ms intervals were applied using a pulse generator (ECM830, BTX) to electroporate the plasmids into neural progenitor cells. Electrodes were positioned to target expression in the mPFC. The embryos were returned to the abdominal cavity and allowed to continue with development. At P0, mice were screened under a fluorescence stereomicroscope to ensure electroporation success, as defined by fluorescence in the targeted brain region, before proceeding with downstream experiments.

## Brain fixation and immunofluorescent staining

P7-9 (neonates) and P28-43 (young adults) mice were deeply anesthetized with isoflurane and sodium pentobarbital (85 mg/kg i.p. injection) and perfused with ice-cold phosphate-buffered saline (PBS) followed by ice-cold 4% PFA. Whole brains were dissected and post-fixed in 4% PFA for 2 hr and then cryoprotected in 30% sucrose for 24–48 hr at 4 °C until they sank to the bottom of the tubes. Brains were serially cut into 50 μm-thick coronal sections using a freezing microtome and stored in PBS +0.05% sodium azide at 4 °C until use.

For immunofluorescence staining, free-floating brain sections were washed in PBS +0.1% triton X-100 (PBS-T) for 2 × 10 min and permeabilized in PBS +0.3% triton X-100 for 20–30 min. Sections were then incubated in blocking buffer (10% normal goat serum+0.3% BSA+0.3% triton X-100 in PBS) for 1 hr at room temperature and in primary antibodies [anti-p-S6 S240/244 (Cell Signaling

Technology #5364, 1:1000) or anti-HCN4 (Alomone Labs APC-052, 1:500), diluted in 5% normal goat serum+0.3% BSA+0.1% triton X-100 in PBS] for 2 days at 4 °C. Sections were then washed in PBS-T for 3 × 10 min, incubated in secondary antibodies [goat anti-rabbit IgG Alexa Fluor Plus 647 (Invitrogen #A32733, 1:500)] for 2 hr at room temperature, and again washed in PBS-T for 3 × 10 min. All sections were mounted onto microscope glass slides and coverslipped with ProLong Diamond Antifade Mountant (Invitrogen) for imaging. The specificity of the HCN4 antibodies was previously validated in our lab (*Hsieh et al., 2020*). Additionally, no primary antibody control was included to confirm the specificity of the secondary antibodies (*Figure 3—figure supplement 2b*).

## Confocal microscopy and image analysis

Images were acquired using a Zeiss LSM 880 confocal microscope. All image analyses were done using ImageJ software (NIH) and were performed by an investigator blinded to experimental groups. Data were quantified using grayscale images of single optical sections. Representative images were prepared using Adobe Photoshop CC. All quantified images meant for direct comparison were taken with the same settings and uniformly processed.

P28-43 neuron soma size and p-S6 staining intensity were quantified from 20 x magnification images of p-S6 stained brain sections by tracing the soma of randomly selected tdTomato+ cells and measuring the area and p-S6 intensity (mean gray value) within the same cell, respectively. HCN4 staining intensity was quantified from 20 x magnification images by tracing the soma of randomly selected tdTomato+ cells and measuring HCN4 intensity (mean gray value). Staining intensities were normalized to the mean control. 15 cells from two brain sections per animal were analyzed for each of the parameters. Neuron positioning (% cells in layer 2/3) was quantified by counting the number of tdTomato+ cells within an 800 μm × 800 μm region of interest (ROI) on the electroporated cortex. Cells within 300 μm from the pial surface were considered correctly located in layer 2/3 whereas cells outside that boundary were considered misplaced (*Nguyen et al., 2022*; *Hsieh et al., 2016*; *Nguyen et al., 2019*). The distribution of neurons in the cortex was further quantified by dividing the 800 μm × 800 μm ROI into 10 evenly spaced bins (bin width = 80 μm) parallel to the pial surface and counting the number of tdTomato+ cells in each bin. Only cells within the gray matter of the cortex were quantified. Data are shown as % of total tdTomato+ cell count. One brain section per animal was analyzed. P7-9 neuron soma size (figure supplement) was quantified from 10 x magnification images of unstained brain sections by tracing the soma of randomly selected tdTomato+ cells and measuring the area. 30 cells from two sections per animal were analyzed.

## Acute brain slice preparation, patch clamp recording, and analysis

Electrophysiology experiments were performed by an investigator blinded to experimental groups. P26-P51 mice were deeply anesthetized by carbon dioxide inhalation and sacrificed by decapitation. Whole brains were rapidly removed and immersed in ice-cold oxygenated (95% $O_2$/5%$CO_2$) high-sucrose cutting solution (in mM: 213 sucrose, 2.6 KCl, 1.25 $NaH_2PO_4$, 3 $MgSO_4$, 26 $NaHCO_3$, 10 Dextrose, 1 $CaCl_2$, 0.4 ascorbate, 4 Na-Lactate, 2 Na-Pyruvate, pH 7.4 with NaOH). 300 μm-thick coronal brain slices containing the mPFC were cut using a vibratome (Leica VT1000) and allowed to recover in a holding chamber with oxygenated artificial cerebrospinal fluid (aCSF, in mM: 118 NaCl, 3 KCl, 1.25 $NaH_2PO_4$, 1 $MgSO_4$, 26 $NaHCO_3$, 10 Dextrose, 2 $CaCl_2$, 0.4 ascorbate, 4 Na-Lactate, 2 Na-Pyruvate, 300 mOsm/kg, pH 7.4 with NaOH) for 30 min at 32 °C before returning to room temperature (22 °C) where they were kept for 6–8 hr during the experiment.

Whole-cell current- and voltage-clamp recordings were performed in a recording chamber at room temperature using pulled borosilicate glass pipettes (4–7 MΩ resistance, Sutter Instrument) filled with internal solution (in mM: 125 K-gluconate, 4 KCl, 10 HEPES, 1 EGTA, 0.2 $CaCl_2$, 10 di-tris-phosphocreatine, 4 Mg-ATP, 0.3 Na-GTP, 280 mOsm/kg, pH 7.3 with KOH). Fluorescent (i.e. electroporated) neurons in the mPFC were visualized using epifluorescence on an Olympus BX51WI microscope with a 40 X water immersion objective. Recordings were performed on neurons in layer 2/3. Data were acquired using Axopatch 200B amplifier and pClamp 10 software (Molecular Devices), and filtered (at 5 kHz) and digitized using Digidata 1320 (Molecular Devices). All data analysis was performed offline using pClamp 10 software (Molecular Devices) and exported to GraphPad Prism 9 software for graphing and statistical analysis.

The RMP was recorded within the first 10 s after achieving whole-cell configuration at 0 pA in current-clamp mode. The membrane capacitance was measured in voltage-clamp mode and calculated by dividing the average membrane time constant by the average input resistance obtained from the current response to a 500 ms long (±5 mV) voltage step from –70 mV holding potential. The membrane time constant was determined from the biexponential curve best fitting the decay phase of the current; the slower component of the biexponential fit was used for the membrane time constant. The resting membrane conductance was measured in current-clamp mode and calculated using the membrane potential change induced by –500 pA hyperpolarizing current injections from rest. The AP input-output curve was generated by injecting 500 ms-long depolarizing currents steps from 0 to 400 pA in 20 pA increments from the RMP of each condition in current-clamp mode. The number of elicited APs was counted using the threshold search algorithm in Clampfit (pClamp 10). To determine the minimum amount of current needed to induce the first AP, i.e., rheobase, 5 ms-long depolarizing current steps in increments of 20 pA were applied every 3 s until an AP was elicited. The first ISI was calculated by measuring the time between the peaks of the first and second AP spike in a trace of ≥10 spikes. The AP threshold, peak amplitude, and half-widths were analyzed from averaged traces of 5-10 consecutive APs induced by the rheobase +10 pA. The AP threshold was defined as the membrane potential at which the first derivative of an evoked AP achieved 10% of its peak velocity (dV/dt). The AP peak amplitude was defined as the difference between the peak and baseline. The AP half-width was defined as the duration of the AP at the voltage halfway between the peak and baseline. $I_h$ was evoked by a series of 3 s-long hyperpolarizing voltage steps from –120 mV to –40 mV in 10 mV increments., with a holding potential of –70 mV. The $I_h$ amplitudes ($\Delta I$) were measured as the difference between the instantaneous current immediately following each test potential ($I_{inst}$) and the steady-state current at the end of each test potential ($I_{ss}$) (*Thoby-Brisson et al., 2000*).

Zatebradine (40 µM, Toris Bioscience) and $BaCl_2$ (200 µM, Sigma-Aldrich) were applied locally to the recorded neurons via a large-tip (350 µM diameter) flow pipe. When no drugs were applied, a continuous flow of aCSF was supplied from the flow pipe. The IV curve, ΔIV curve, and RMP were measured before and after drug application as described above. The zatebradine-sensitive and $BaCl_2$-sensitive currents were obtained by subtracting the current traces obtained after from before the drug application. The IV curve for the zatebradine-sensitive current was obtained by measuring the steady-state of the resulting current, and the IV curve for the $BaCl_2$-sensitive current was obtained by measuring the peak of the resulting current.

sEPSCs were recorded at a holding potential of –70 mV. Synaptic currents were recorded for a period of 2–5 min and analyzed by using the template search algorithm in Clampfit (pClamp 10). The template was constructed by averaging 5–10 synaptic events, and the template match threshold parameter was adjusted to minimize false positives. All synaptic events identified by the program were manually inspected and non-synaptic currents (based on the fast-rising time) were discarded. The total charge (pA/ms) was calculated as the area of sEPSC events (pA/ms) × frequency (Hz)/1000 for each neuron.

## Statistical analysis

All statistical analyses were performed using GraphPad Prism 9 software with the significance level set at $p < 0.05$. Data distribution was assumed to be normal. Data were analyzed using nested t-test or nested one-way ANOVA obtained by fitting a mixed-effects model wherein the main factor is treated as a fixed factor and the nested factor is treated as a random factor; to account for correlated data within individual animals within groups (*Moen et al., 2016*; *Yu et al., 2022*), one-way ANOVA, two-way repeated measure ANOVA, or mixed-effects ANOVA (fitted to a mixed-effects model for when values were missing values in repeated measures analyses), as appropriate. For all nested statistics, the distribution of data among individual animals is shown in the accompanying figure supplements. All post-hoc analyses were performed using Holm-Šídák's multiple comparison test. The specific tests applied for each dataset, test results, and sample size (n, number of animals or neurons) are summarized in the accompanying source data and described in the figure legends. All data are reported as the mean of all neurons or brain sections in each group ± standard error of the mean (SEM).

## Materials availability

Materials (plasmids) generated as part of this study are available upon reasonable request to the corresponding authors.

## Acknowledgements

We thank Drs. Stéphanie Baulac (Paris Brain Institute, France) for providing the pX330-*Depdc5* plasmid and Dr. Joseph LoTurco (University of Connecticut) for insightful discussions and advice on the pX330-*Tsc1* plasmid.

## Additional information

### Competing interests

Angelique Bordey: AB is an inventor on a patent application "Methods of treating and diagnosing epilepsy" (Pub. no. US2022/0143219 A1). The other authors declare that no competing interests exist.

### Funding

| Funder | Grant reference number | Author |
| --- | --- | --- |
| National Institutes of Health | F32 HD095567 | Lena H Nguyen |
| National Institutes of Health | R01 NS111980 | Angelique Bordey |

The funders had no role in study design, data collection and interpretation, or the decision to submit the work for publication.

### Author contributions

Lena H Nguyen, Conceptualization, Formal analysis, Investigation, Visualization, Writing – original draft, Writing – review and editing; Youfen Xu, Formal analysis, Investigation; Maanasi Nair, Formal analysis; Angelique Bordey, Conceptualization, Writing – original draft, Writing – review and editing

### Author ORCIDs

Lena H Nguyen (ORCID) https://orcid.org/0000-0002-4463-7164
Angelique Bordey (ORCID) https://orcid.org/0000-0003-3496-3385

### Ethics

This study was performed in strict accordance with the recommendations in the Guide for the Care and Use of Laboratory Animals of the National Institutes of Health. All of the animals were handled according to approved Institutional Animal Care and Use Committee (IACUC) protocols of Yale University (protocol number: 2022-10514).

Reviewer #1 (Public Review): https://doi.org/10.7554/eLife.91010.3.sa1
Reviewer #2 (Public Review): https://doi.org/10.7554/eLife.91010.3.sa2
Author Response https://doi.org/10.7554/eLife.91010.3.sa3

## Additional files

### Supplementary files

• MDAR checklist

### Data availability

All data generated and analyzed during this study are included in the manuscript and supporting files.

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
