## [Editor Report · eLife assessment]

This manuscript examines shared and divergent mechanisms of disruptions of five different mTOR pathway genes on embryonic mouse brain neuronal development. The significance of the manuscript is **important**, because it bridges several different genetic causes of focal malformations of cortical development. The strength of evidence is **compelling**, relying on both gain and loss of function, demonstrating differential impact on excitatory synaptic activity, conferring gene-specific mechanisms of hyperexcitability. The results have both theoretical and practical implications for the field of developmental neurobiology and clinical epilepsy.

---

## [Referee Report · Reviewer #1 (Public Review)]

Pathogenic mutations of mTOR pathway genes have been identified in patients with malformation of cortical development and intractable epilepsy. Nguyen et al., established an in vivo rodent model to investigate the impact of different mTOR pathway gene dysfunction on neuronal intrinsic membrane excitability and cortical network activity. The results demonstrate that activation of mTORC1 activators or inactivation of mTORC1 repressors leads to convergent mTOR pathway activation and alterations of neuronal morphology, the key pathological feature of human FCD and hemimegalencephaly. However, different mTOR pathway gene mutations also exhibited variations in modulating Ih current and synaptic activity in rodent cortical neurons. These findings provide novel insights into the mechanism of seizure generation associated with cortical malformation.

---

## [Referee Report · Reviewer #2 (Public Review)]

Summary:

The study provides valuable and compelling evidence that while activation of the mTOR cascade confers some similarities in alterations in cell size, mTOR pathway activation, cortical lamination, baseline firing properties, and synaptic activity, there are distinctions that could account for clinical differences in seizure and epilepsy phenotypes in patients harboring these mutations. These findings could have important implications going forward as we design clinical therapeutic strategies to modulate mTOR activity in these individuals to treat seizures.

This study presents a valuable finding on the role that distinct mTOR pathway genes play in altered cell shape, cortical laminar migration, and cellular excitability in the mouse medial prefrontal cortex (mPFC). The evidence supporting the claims of the authors is solid, although analysis of the role of the mTORC2 pathway and consideration of distinct metabolic states i.e., amino acid levels would have strengthened the study. The work will be of interest to neuroscientists working on human epilepsy. These genes have each been assayed in previous independent studies and thus the direct comparison is what provides the innovation and interest.

The manuscript by Nguyen and colleagues attempts to define both the common and differential roles of mTOR pathway genes, both by gene knockout (KO) and activation, on cortical neuronal size, cortical lamination, and excitability. They focused on 5 genes that have been linked to human malformations of cortical development (MCD) and epilepsy: RhebY35L, mTORS2215Y, Dedpdc5KO, PtenKO, and Tsc1KO. The RhebY35L, mTORS2215Y are known and pathogenic human gain-of-function variants. Each of these genes is known to modulate the activity of mTORC1 and either KO or activation will lead to abnormal and persistent hyperactivation of mTOR activity. Using in utero electroporation they transfected plasmids containing these gene constructs into fetal mouse brains at E15.5 and then assessed neuronal shape and size, laminar positioning, spontaneous activity, synaptic activity, and expression of a novel voltage-gated potassium channel (HCN4) at varying time postnatally e.g., P7-9 (neonates) and P28-43 (young adults).

The study clearly achieves its stated aims i.e., that disruption of each of five distinct mTOR pathway genes, Rheb, mTOR, Depdc5, Pten, and Tsc1, individually impacts pyramidal neuron development and electrophysiological function in the mouse mPFC. The data from each of the 5 genes provides strong support to the notion that mTOR pathway gene mutations yield the unifying clinical parcellation of mTORopathies, likely as a consequence of mTOR pathway activation. The data also provide interesting evidence that subtle or even overt differences in clinical phenotypes between RhebY35L, mTORS2215Y, Dedpdc5KO, PtenKO, and Tsc1KO in humans could be due to effects of these genes either on mTOR or on yet to be defined alternative pathways. Assuredly follow-up studies to examine how Rheb, mTOR, Dedpdc5, Pten, and Tsc1 engage with other protein binding partners or other pathways will be warranted in future studies.

Strengths:

The investigators demonstrate that gene KO or activation leads to common changes in cell size (enlargement) though with different effects across each gene subtype suggesting distinct genetic effects despite a common effect on mTOR signaling. The major effect was seen in forebrain neurons expressing mTORS2215Y. They also report gene-specific effects of each mTOR pathway gene on cortical lamination. For example, while RhebY35L, mTORS2215Y, Dedpdc5KO, and Tsc1KO significantly disrupted laminar positioning of neurons in layer 2/3, PtenKO had minimal effects on laminar positioning. This finding is intriguing since it means that simply activating mTOR during fetal brain development will not necessarily alter cortical lamination and that an increase in cell size by itself doesn't disrupt laminar fidelity. To verify that the expression of plasmids led to mTORC1 hyperactivation, phosphorylated levels of S6 (i.e., p-S6), a downstream substrate of mTORC1, were assayed by immunohistochemistry in P28-43 mice. Expression of the RhebY35L, mTORS2215Y, Dedpdc5KO, PtenKO, and Tsc1KO plasmids all led to significantly increased p-S6 staining intensity, supporting that the expression of each of these plasmids leads to increased mTORC1 signaling.

Whole-cell current- and voltage-clamp recordings were performed in P25-P51 mice in acute brain slice preparations. Expression of RhebY35L, mTORS2215Y, Dedpdc5KO, PtenKO, and Tsc1KO led to decreased depolarization-induced excitability, but only RhebY35L, mTORS2215Y, and Tsc1KO expression led to depolarized resting membrane potentials. Interestingly, expression of RhebY35L, mTORS2215Y, Dedpdc5KO, PtenKO, and Tsc1KO led to the abnormal presence of HCN4 channels with variations in functional expression suggesting a common cellular mechanism that could confer excitability. Treatment with rapamycin, an mTOR inhibitor, reversed the expression changes in HCN4. Expression of RhebY35L, mTORS2215Y, Dedpdc5KO, PtenKO, and Tsc1KO led to different impacts on sEPSC properties. Effects of treatment with the selective HCN channel blocker zatebradine on hyperpolarization-induced inward currents in mTORS2215Y neurons confirmed the identity of ΔI as Ih.

Overall the data presented provides a convincing and compelling direct comparison of the roles that select mTOR pathway genes play on brain development and network excitability. It is critical to directly compare these gene effects in mouse models because although these genes are part of the mTOR pathway and clearly cause augmentation of mTOR activation, there are mechanistic differences in how these gees modify mTOR and how they interact with other proteins and phenotypic differences in humans harboring mutations in these same genes.

---

## [Author Response]

The following is the authors’ response to the original reviews.

**Public Reviews:**

We thank the reviewers for their insightful comments on our manuscript. We have addressed the reviewers’ comments below and in the revised manuscript.

Reviewer #1:Comment #1: The authors found differences in the initial spike doublet of action potentials between cortical neurons in experimental and control conditions (Figure 2e). The action potential firing frequency of the first two APs (instant firing frequency) of recorded neurons shall be quantified to investigate whether there are statistical differences between the action potential firing frequency in cortical neurons in different experimental groups versus control conditions.

Response: As suggested by the reviewer, we have quantified the first interspike interval (ISI; time between the 1st and 2nd action potential). The data is included in Figure 2h as well as in Figure 2—figure supplement 1e and Figure 2—source data 1. The Results and Methods have also been updated accordingly.

Comment #2: The mTORS12215Y induced the largest changes in Ih current amplitudes in cortical neurons compared with other experimental conditions. Whether the HCN4 channel expression is regulated by mTOR pathway activation, or could there be possible interactions between the HCN channel and mTORS12215Y mutant protein?

Response: Our previous findings using the RhebS16H mutation support the idea that increased expression of HCN4 channels is regulated by mTOR pathway activation. This is evidenced by its sensitivity to rapamycin (a mTOR inhibitor) and expression of constitutively active 4E-BP1 (a translational repressor downstream of mTORC1). Since mTORS2215Y directly hyperactivates mTORC1 and there are no known interactions between HCN channels and mTORS2215Y, our data strongly suggests that abnormal HCN4 channel expression occurs via mTORC1 hyperactivation in this condition. We have revised our Discussion to make this point clearer.

Comment #3: A comparison of the electrophysiological characteristics of cortical neurons in different experimental conditions in the present study and pathological neurons in human FCD reported in previous literature could be interesting. Inducing pathological gene mutations or knocking out key genes in mTOR pathway in the rodent cortex - which approach could better model human FCD?

Response: We agree with the reviewer and have added a new paragraph in the Discussion to compare our electrophysiology results to those of previous studies done on human FCDII and TSC cytomegalic neurons. With regards to the reviewer’s question about which of the two approaches in the rodent cortex – inducing pathological gene mutations or knocking out key genes in the mTOR pathway – would better model human FCD, our study emphasizes the importance of considering gene-specific mechanisms in FCDII. Thus, modeling the genetically distinct FCDIIs will require using gene-specific manipulations. We have revised our Discussion to include this point. With that said, for some phenotypes that are generalized across FCDII independent of the mTOR pathway genes, using pathogenic mutations of mTOR activators or knockout of negative mTOR regulators would likely both be appropriate models. Of note, as discussed in the manuscript, there are also technical factors to be considered when choosing to use a pathogenic gene mutation versus knocking out a gene (the latter which would depend on the half-life of the proteins).

**Reviewer #2:**
Comment #1: The authors postulate that all the findings are dependent on mTORC1-related effects but don't assess whether some of the differences could be due to effects on mTORC2 signaling. mTORC2 is an important and poorly understood alternative isoform of mTOR (due to rictor binding) that has effects on distinct cell signaling pathways and in particular actin polymerization. This doesn't diminish the effects of the current analysis of mTORC1 but could explain genotypic differences in each variable. A few prior studies have assessed the role of mTORC2 in epileptogenesis and cortical malformations (Chen et al., 2019).

Response: We agree with the reviewer and have revised our Discussion to include the possibility of mTORC2 contribution to the gene-specific phenotypic differences.

Comment #2: The slice recordings were performed in the usual recording aCSF buffer conditions but there is no assessment of the role of amino acids or nutrients in the bath. While it is clear that valuable and viable acute slice recordings can be made in aCSF, the role of the mTOR pathway is to modulate cell growth in response to nutrient conditions. Thus, one variable that could be manipulated and assessed currently in this study is the levels of amino acids i.e., leucine and arginine added to the bath since DEPDC5 and TSC1 are responsive to ambient amino acid levels.

Response: We thank the reviewer for this great suggestion, and we intend to pursue this as part of another study.

Comment #3: The analysis concedes that the role of somatic mutations in cortical malformations may depend not only on genotypic effects but also on allelic load and cellular subtype affected by the mutation. Thus, it would be interesting to see if electroporation either at E14 or E16, thereby affecting a distinct pool of progenitors, would mitigate or accentuate differences between mTOR pathway genes.

Response: We agree with the reviewer. This is a crucial experiment that we hope to perform in the future. We have also added a paragraph in our Discussion to address this important point.

Comment #4: Treatment with rapamycin and zatebradine in each condition would have added to the strength of the findings to determine the mTOR-dependence and reversibility of HCN4 effects.

Response: We previously demonstrated the mTORC1 dependence of HCN4 expression in the RhebS16H condition using rapamycin and expression of constitutively active 4E-BP1. 4E-BP1 is a translational repressor downstream of mTORC1. In the 4E-BP1 study, we used a conditional system to express 4EBP1F113A (mutation that resists inactivation by mTORC1) in adolescent mice while RhebS16H (and thus mTORC1 activation) was expressed embryonically. 4E-BP1F113A expression suppressed Ih current and HCN4 expression, suggesting that aberrant HCN4 expression can be reversed by decreasing mTORC1regulated translation. Based on these data and the findings that rapamycin suppressed abnormal HCN4 expression, we postulate that increased HCN4 expression in the different gene conditions examined in the present study occurs via the mTORC1 pathway. However, we agree with the reviewer that treating each of the conditions with rapamycin would provide direct evidence of their mTORC1 dependence. Additionally, treating each condition with the HCN channel blocker zatebradine would also add strength to the findings. We have added a comment in the Discussion to acknowledge this point.

**Reviewer #1 (Recommendations For The Authors):**
Comment #1: The authors found increased frequency or amplitudes of spontaneous postsynaptic currents in different experimental cohorts. These data may not be sufficient to conclude increased synaptic excitability, because there are no pharmacological experiments to verify whether the recorded inward currents are excitatory or inhibitory postsynaptic currents. An alternative approach could be analyzing the decay time of spontaneous postsynaptic currents, the excitatory postsynaptic currents had relatively faster decay time compared with inhibitory postsynaptic currents.

Response: Thank you for the comment. We apologize for the lack of clarity and have added the following text in the Results to clarify: “To separate sEPSCs from spontaneous inhibitory postsynaptic currents (sIPSCs), we used an intracellular solution rich in K-gluconate to impose a low intracellular Cl- concentration and recorded at a holding potential of -70 mV, which is near the Cl- reversal potential. The 90%-10% decay time of the measured synaptic currents ranged between 4-8 ms in all conditions (mean ± SD: control: 4.9 ± 1.6; RhebY35L: 5.2 ± 1.4; mTORS2215Y: 7.4 ± 1.4; control: 6.8 ± 0.7; Depdc5KO: 7.4 ± 1.0; PtenKO: 8.1 ± 0.9; Tsc1KO: 7.4 ± 0.9), consistent with the expected decay time for sEPSCs and shorter than the decay time for sIPSCs (Kroon et al, 2019). The recorded synaptic currents were therefore considered to be sEPSCs.”

Comment #2: There are typos of Depdc5 in the text and figure legends.

Response: Thank you for noticing this error. We have corrected the typos in the manuscript.